# A Multinational Pilot Study on Patients’ Perceptions of Advanced Neuroendocrine Neoplasms on the EORTC QLQ-C30 and EORTC QLQ-GINET21 Questionnaires

**DOI:** 10.3390/jcm11051271

**Published:** 2022-02-25

**Authors:** Rachel S. van Leeuwaarde, Angélica M. González-Clavijo, Marc Pracht, Galina Emelianova, Winson Y. Cheung, Christina Thirlwell, Kjell Öberg, Francesca Spada

**Affiliations:** 1Department of Endocrine Oncology, University Medical Center Utrecht, 3584 Utrecht, The Netherlands; r.vanleeuwaarde@umcutrecht.nl; 2Department of Physiological Sciences, School of Medicine, Universidad Nacional de Colombia, Bogota 111321, Colombia; angelik_md@yahoo.com; 3Instituto Nacional de Cancerología, Bogota 111321, Colombia; 4Department of Medical Oncology, Centre Eugène Marquis, 35000 Rennes, France; m.pracht@rennes.unicancer.fr; 5Department of Oncology, National Medical Research Center N.N. Blokhin, 115191 Moscow, Russia; docgalina@mail.ru; 6Department of Medicine and Dentistry, A.I. Yevdokimov Moscow State University, 127473 Moscow, Russia; 7Department of Oncology, University of Calgary, Tom Baker Cancer Center, Calgary, AB T2N 4N2, Canada; winson.cheung@ubc.ca; 8Cancer Institute, University College London, London WC1E 6DD, UK; christina.thirlwell@ucl.ac.uk; 9Department of Medicine and Health, University of Exeter School, Exeter EX4 4PY, UK; 10Department of Endocrine Oncology, Uppsala University Hospital, 75185 Uppsala, Sweden; kjell.oberg@medsci.uu.se; 11Department of Medical Sciences, Uppsala University, 75236 Uppsala, Sweden; 12Division of Gastrointestinal Medical Oncology and Neuroendocrine Tumors, European Institute of Oncology (IEO), Istituto di Ricovero e Cura a Carattere Scientifico, 20141 Milan, Italy

**Keywords:** neuroendocrine tumors, NETs, quality of life, EORTC QLQ-C30, EORTC QLQ-G.I.NET21

## Abstract

Among the available neuroendocrine neoplasm (NEN)-specific HR-QoL scales, only the EORTC QLQ-C30 and EORTC QLQ-G.I.NET21 questionnaires have been validated in several languages. We aim to assess patients’ perceptions of these questionnaires. A cross-sectional qualitative pilot study was conducted among 65 adults from four countries with well-differentiated advanced gastro-entero-pancreatic (GEP) or unknown primary NENs. Patients completed the EORTC QLQ-C30 and EORTC QLQ-G.I.NET21 questionnaires and then a survey containing statements concerning the questionnaires. The majority of patients had a small intestine NET (52%). Most tumors were functioning (55%) and grade 2 NET (52%). Almost half of the patients identified limitations in the questionnaires, with nine (14%) patients scoring the questionnaires as poor and 16 (25%) patients as moderate. Overall, 37 (57%) patients were positive towards the questionnaires. Approximately a quarter of patients considered the questionnaires not suitable for all ages, missing some of their complaints, not representative of their overall HR-QoL regarding the treatment of their NET and too superficial. The current validated EORTC QLQ-C30 and EORTC QLQ-G.I.NET21 questionnaires may show some limitations in the design of questions and the patients’ final satisfaction reporting of the questionnaire. Large-scale, high-quality prospective studies are required in HR-QoL assessment regarding NETs.

## 1. Introduction

Neuroendocrine neoplasms (NENs) constitute a heterogeneous group of tumors with increasing incidence worldwide [1]. Most tumors are clinically silent until a late stage, when symptoms emerge due to the tumors’ mass effects. Characteristically, NENs can secrete a variety of neuroamines and hormonal polypeptides [2], often leading to distinct clinical syndromes, including carcinoid syndrome, which is frequently referred to as functioning NENs. Some NENs are associated with a genetic syndrome, but most are sporadic [3].

Gastro-entero-pancreatic (GEP) NENs are classified as grade (G) 1, G2 or G3 in accordance with the 2019 WHO classification for GEP-NENs [4]. Well-differentiated G1, G2 and G3 lesions are called neuroendocrine tumors (NETs), whereas poorly differentiated G3 NENs are described as neuroendocrine carcinomas (NECs) [4].

While surgery is the only curative treatment in patients with localized NETs, several systemic options, including chemotherapy, somatostatin analogs, IFN-α 2b, peptide receptor radionuclide therapy (PRRT), and molecular targeted agents, as well as liver directed therapies, are available in patients with advanced unresectable or metastatic disease.

Given the biological heterogeneity and the improvement in therapeutic options, NET patients can live for many years with advanced disease. Therefore, treatment goals in these patients include improving survival, managing symptoms, and controlling tumor growth. Considering long-term survival, preservation of health-related quality of life (HR-QoL) has become a major priority [5].

Whilst many previous studies have evaluated the impact of these therapies on survival [6,7,8,9,10,11,12,13,14,15,16], only a few studies [13,17,18,19,20,21,22,23] have reported the impact of the disease and its subsequent treatment on HR-QoL. Patients with NETs have demonstrated a worse HR-QoL compared to that of the general population [24]. Several questionnaires have been used to evaluate HR-QoL in patients with NETs. The most widely used instrument for measuring HR-QoL is the European Organization for Research and Treatment of Cancer Quality-of-Life Questionnaire (EORTC QLQ-C30) [25], a generic scale that does not cover specific NET-related problems or domains. The EORTC QLQ-G.I.NET21 has been designed especially for patients with NETs [26]. It has been validated in multinational studies [27,28] and translated into nine languages according to the EORTC translation guidelines [29]. It is considered an accurate tool in assessing topics such as toxicities, symptoms, and NET-related tumor progression compared to the EORTC QLQ-C30 tool [26].

The purpose of this study was to assess patients’ attitudes and perceptions of the EORTC QLQ-C30 and EORTC QLQ-G.I.NET21 scales, identify aspects of the questionnaires that may be improved and evaluate the HR-QoL in the different patient populations.

## 2. Methods

### 2.1. Study Population

The following inclusion criteria were used: >18 years old; Eastern Cooperative Group (ECOG) performance status (PS) ≤ 2; confirmation of a histological diagnosis of well-differentiated GEP, namely G1, G2, G3 NETs in accordance with 2019 WHO classification(4); or well-differentiated unknown primary NEN at the enrolling centers and locally advanced/metastatic tumor stage.

Demographic, clinical and biological information were retrieved from the database of each institution during the timeframe of September 2008 to March 2018.

Participants were recruited from the University Medical Center Utrecht (Netherlands), the Cancer Institute Eugène Marquis in Rennes (France), the National Medical Center of Oncology N.N. Blokhin in Moscow (The Russian Federation) and the National Cancer Institute in Bogotá (Colombia). Recruitment used consecutive sampling with patients visiting the outward patient clinic or attending the clinic for treatment with radioembolization or PRRT.

### 2.2. Study Design

A cross-sectional qualitative pilot study was conducted prospectively among eligible patients who completed the EORTC QLQ-C30 and EORTC QLQ-G.I.NET21 questionnaires from January 2017 until December 2017 in the recruiting centers. The questionnaires were completed by hand simultaneously by patients in all countries.

All participants provided written informed consent. The Medical Ethical Committee of the University Medical Center of Utrecht confirmed that the Medical Research Involving Human Subjects Act (WMO) did not apply for this study. Therefore, official approval was not required under WMO.

### 2.3. Questionnaires

Patients were invited to complete the questionnaire. To avoid potential biases resulting from health literacy, the researchers were available to clarify all questions while the patients filled out the surveys. We used the EORTC QoL scoring manual instructions for domains with multiple questions; if ≥50% but ≤100% of questions were completed, then they were deemed evaluable for that domain, and the average of the remaining assessed questions was used in the analysis. If ≥50% of questions were missing for a particular domain, they were excluded from the analysis. As per the EORTC manual questionnaire, results were normalized to a 100-point scale. In the EORTC QLQ-C30 questionnaire, the Global Health Status and five Function Scale domains (physical, role, emotional, cognitive, and social) are positive scales in which higher scores translate into higher HRQoL. Symptom scales are negative scales in which a higher score corresponds to a higher level of symptoms/problems. These include fatigue, nausea and vomiting, pain, dyspnea, insomnia, appetite loss, constipation, diarrhea, and financial difficulties. The EORTC QLQ-G.I.NET21 domains include an endocrine scale (flushing, sweats), a GI scale (bloating, flatulence), a treatment scale, a social functioning scale, a disease-related worries scale, muscle/bone pain, sexual function, information/communication function, and body image. All are negative scales in which higher scores correspond to increased symptoms/problems.

After completing the EORTC QLQ-C30 and EORTC QLQ-G.I.NET21 questionnaires, patients completed a survey regarding the questionnaires. This survey was arbitrarily designed specifically for this study (Table 1) by the authors based on the most frequent questions from the patients during routine clinical practice. It contained ten statements regarding the EORTC QLQ-C30 and EORTC QLQ-G.I.NET21 questionnaires. An agreement score for each item of this survey was defined. The total scores were classified as “poor,” “moderate,” or “good,” and they were defined as total scores of one to five, six to seven or eight to nine, respectively. A higher score correlated with a positive attitude towards the EORTC QLQ-C30 and EORTC QLQ-G.I.NET21 questionnaires.

### 2.4. Statistical Analysis

General descriptive statistics were applied to characterize the study population. Analyses were conducted using SPSS 22.0.

The explorative nature of the study did not require a specific sample size; however, the aim was to have a minimum of 50 participants.

## 3. Results

### 3.1. Population Characteristics

A total of 65 patients were included in this study. The mean age of the study population was 57 (SD ±12.7) years. Patients had a GEP-NET diagnosis (92%) or unknown primary NET (8%). Most had a midgut NET (52%) and a functioning tumor (55%). Most patients had a G2 NET (45%). Half of the whole population was from The Netherlands (Table 2). The current treatments of the whole population at the moment of the survey are summarized in Table 3.

The majority of patients were previously treated with somatostatin analogs (9%), followed by PRRT (26%), radio-embolization (17%), chemotherapy (14%), mTOR inhibitor (11%), and tyrosin–kinase inhibitor (6%).

### 3.2. Patients’ Satisfaction Regarding the EORTC QLQ-C30 and G.I.NET-21 Questionnaires

Almost half of the patients identified limitations in the EORTC QLQ-C30 and EORTC QLQ-G.I.NET21 questionnaires. Moreover, approximately a quarter of patients considered the questionnaires unsuitable for all ages, did not address all their complaints, did not represent their overall HR-QoL regarding the treatment of their NET, and found them too superficial. About 16% preferred to complete the questionnaires with the help of their physician (Table 4).

### 3.3. Health-Related Quality of Life

The outcomes of the EORTC QLQ-C30 and EORTC QLQ-G.I.NET21 scores are presented as both mean and median scores to give insight into the distribution of the data. The HR-QoL scores of the whole population are reported in Table 5.

#### 3.3.1. Current Treatment

The majority of patients were treated with somatostatin analogs (71%), followed by PRRT (22%), radio-embolization (20%), a mTOR inhibitor (3%) or chemotherapy (11%).

#### 3.3.2. Comments from Patients on the HR-QoL Questionnaires

The main complaints on both questionnaires included some specific medical terms considered difficult to understand and the lack of questions judged relevant in the patients’ perception (Table 6).

#### 3.3.3. Characteristics of Patients with the Lowest Scores

Among the whole population, a total of nine (14%) patients scored the questionnaires as “poor,” whereas a total of 56 patients scored the questionnaires as “moderate” or “good”. The questionnaires were judged “poor” mostly by males rather than females.

A total of 21/56 (38%) patients who scored the questionnaire as “moderate” or “good” were currently on PRRT or radioembolization, compared to 3/9 (33%) patients who scored the questionnaire as “poor.” The distribution of patients in somatostatin analogs, age, the functionality of the tumor and grade were equally presented in the “poor” and “moderate”/“good” scoring groups.

## 4. Discussion

This prospective, multicenter, multinational, real-life pilot study suggests that, based on the patients’ perceptions, the EORTC QLQ-C30 and EORTC QLQ-G.I.NET21 questionnaires are not completely representative of all the aspects of the patients’ HR-QoL during the treatment of their NET.

Half of the patients evaluated the EORTC scales as not sufficiently addressing their problems, suggesting that the questionnaires could be better tailored to their requirements.

Clinical studies conducted over the past two decades have demonstrated that patients’ perception of their own HR-QoL and treatment-related aspects have an essential role in the global evaluation of their disease experience. Nevertheless, conventional physician assessment of a patient’s morbidity very often does not correlate with a patient’s self-reported description of their daily functional activities and well-being [30]. These aspects are represented in the concept of HR-QoL. There is agreement that HR-QoL describes “the extent to which one’s usual or expected physical, emotional, and social well-being is affected by a medical condition or its treatment” [31] and needs to be evaluated as a specific outcome that comprises self-report of a patient’s health status without interpretation by a third person [32]. The assessment of HR-QoL as an important secondary endpoint in clinical studies, by using validated self-reported tools, has become a standard criterion in oncology over the last several years [33,34,35], and this high-quality information on HR-QoL could be useful for many reasons, including helping to inform the development of therapeutic interventions to address deficiencies in the QoL [36].

NET patients experience a decreased HR-QoL in comparison with that of the general population [28], whereby functioning tumors are more likely to lead to worse HR-QoL [24]. Moreover, the NET diagnosis has a significant impact not only on the immediate but also on the long term, and the approach to life “after” the NET diagnosis could not be similar to that “before” the NET diagnosis [37,38,39,40].

The importance of determining HR-QoL during different treatment regimens is underscored by recent phase III studies in which the HR-QoL was assessed during PRRT, everolimus and sunitinib [19,20,21]. In particular, the analyses from the NETTER-1 study show that, in addition to improving progression-free survival, 177Lu-Dotatate can enhance QoL, improving the time to deterioration of some parameters such as global health status, physical functioning, role functioning, fatigue, pain, diarrhea, disease-related worries, and body image for patients with progressive midgut NETs compared with high-dose octreotide. The HR-QoL was a secondary end point of this phase III trial and was measured through both EORTC QLQ C-30 and G.I.NET-21 questionnaires. Patients on trial completed these questionnaires at baseline and every 12 weeks until central radiologic confirmation of disease progression during therapy [14,19].

All these parameters are included in the HR-QoL scores of our population, demonstrating that QoL plays a crucial role when evaluating the benefit versus risk of a new cancer treatment. Indeed, in patients with advanced NETs, maintenance of an acceptable or good HR-QoL is particularly relevant, given the relatively long clinical history and overall survival.

However, while the EORTC QLQ-C30 questionnaire is a general tool that can be used to evaluate five functional domains (physical, role, emotional, cognitive, and social) in patients with neoplasms, the EORTC QLQ-G.I.NET21 is a more recognized and valid tool to assess NET-related HR-QoL specifically.

Although both questionnaires are EORTC-validated tools, our survey has highlighted some limitations to their routine use in assessing patients’ HR-QoL. In our real-life survey, we chose to refer specifically to the EORTC QLQ-G.I.NET21 due to its relevance in the NEN field.

A quarter of the patients did not find the questions suitable for all ages. However, few patients missed questions on fertility. Therefore, a dedicated EORTC QLQ-G.I.NET21 questionnaire for adolescents and young adults with NETs could be a proposal to move towards an age-specific questionnaire. Approximately a quarter of patients missed some questions, but this seems to conflict with the feeling of some patients who stated that the questionnaires did not address all HR-QoL-related issues during NET treatment. In particular, some patients highlighted specific aspects that they felt were missing from the questionnaire, including questions related to hair loss and medications, psychological support during treatment, loneliness and solitude, and positive aspects of their life during the disease and treatment course. Almost half of the evaluated patients stated that there should be more emphasis on HR-QoL during the treatment for NET. This seems to be a very high percentage considering that the main goals of most cancer therapeutics are overall survival prolongation.

This study has several limitations, including the small number and heterogeneity of patients enrolled. Neuroendocrine tumors are characterized by the heterogeneity of localizations, hormonal secreting syndromes and proliferation indexes. Subsequently, the clinical manifestations differ from patient to patient. This is also seen in the present study. The EORTC QLQ-G.I.NET21 focuses mostly on carcinoid syndrome and seems less suitable for non-functioning NETs. This can be considered as a pitfall of the study, but also reflects the real-time nature of the study. Future studies that work on the development of sub-questionnaires to address the different types of NETs are warranted.

The limited population size reflects the rarity of the disease and highlights the need for multinational collaboration. A pilot study was undertaken in line with an expected small sample size. Another limitation is the imbalance of selection of the patients among the centers due to different catchment areas. However, the results from this study are comparable with the scores presented in the NETTER-1 study [19]. This suggests that this population is representative regarding the HR-QoL in which the majority of patients were treated with either somatostatin analogs or PRRT. Most patients were enrolled from one center in the Netherlands. This center is a European Neuroendocrine Tumor (ENET) center of excellence with a high referral rate of patients with NENs. Therefore, the number of patients is highest in this center and lower in the smaller expertise centers.

The heterogeneity of the studied population represents a strength with both functioning and non-functioning tumors included, especially since the EORTC QLQ-G.I.NET21 has not been specified for either functioning or non-functioning tumors, so the current population is the target population. Another strength is the multinational design of the study in which patients with different cultural backgrounds were included. This reflects the real-life setting of this study.

In summary, despite an increasing interest in HR-QoL NET patients, there is still a lack of knowledge on HR-QoL over time. Our survey suggests that the valid application of HR-QoL findings in clinical practice is hampered by several limitations, such as the wide use of novel treatment agents that may entail a range of symptoms not currently covered by validated questionnaires, the lack of specific questions on particular domains, the use of the same tools regardless of age and lifestyle, and the gap between the available questionnaires that evaluate different time points in the disease trajectory.

To conclude, our analysis is a hypothesis-generating study suggesting that the current HR-QoL questionnaires seem to show some limitations in the design of questions and the final reporting of the questionnaires. Therefore, large-scale, high-quality prospective studies are required in this field.

## Figures and Tables

**Table 1 jcm-11-01271-t001:** Survey results.

Survey Questions	Answers
Did you understand all the questions?	98%
I think this questionnaire is too long	12%
Completion of this questionnaire is too strenuous	10%
Do you prefer to complete the questionnaire with the help of your physician?	17%
This questionnaire is suitable for all ages	76%
Did this questionnaire address all your complaints?	71%
Did this questionnaire represent my overall quality of life regarding the treatment for my NET?	79%
This questionnaire was too superficial	24%
I missed questions on fertility	10%
There should be more emphasis on quality of life during treatment for NET	60%

NET: Neuroendocrine tumor.

**Table 2 jcm-11-01271-t002:** Baseline characteristics of the whole population of the study.

Baseline Characteristics	*n* = 65, *n* (%)
Age (years), mean ± standard deviation	57 ± 12.8
Gender (male/female)	34/31
Location:	
• Pancreas	19 (29%)
• Gastric	2 (3%)
• Small intestine	34 (52%)
• Caecum	1 (2%)
• Rectum	4 (6%)
• Unknown primary	5 (8%)
Functioning tumor	36 (55%)
Tumor grade	
• Grade 1	23 (35%)
• Grade 2	34 (52%)
• Grade 3	8 (12%)
Country:	
The Netherlands	33 (52%)
France	10 (16%)
Russia	9 (14%)
Colombia	12 (19%)

Tumor grade reflects the 2019 WHO classification.

**Table 3 jcm-11-01271-t003:** Current treatment characteristics of the whole population of the study.

Treatment	*n* (%)
Somatostatin analogs	46 (71%)
PRRT	14 (22%)
Radio-embolization	13(20%)
m-TOR inhibitor	2 (3%)
Chemotherapy	7 (11%)

PRRT = peptide receptor radionuclide therapy.

**Table 4 jcm-11-01271-t004:** HR-QoL scores of the whole population.

HR-Qol ScoresHR-Qol Scores	*n* (%*n* (%))
Poor	9 (14)
Moderate	16 (25)
Good	37 (57)

Poor: total score of 1–5, Moderate: total score of 6–7, Good: total score of 8–9.

**Table 5 jcm-11-01271-t005:** HR-QoL scores of the whole population.

HR-QoL Domain	No.	Mean (SD)	Median (Range)
Global health status/QoL	63	58.9 (23.8)	66.7 (17, 100)
Physical functioning	63	76.4 (22.2)	80 (0, 100)
Role functioning	65	63.8 (31.9)	66.7 (0, 100)
Emotional functioning	61	74.0 (17.9)	75 (25, 100)
Social functioning	63	73.2 (24.2)	83 (33, 100)
Fatigue	63	42.0 (25.7)	33 (0, 100)
Nausea and vomiting	65	17.7 (25.8)	17 (0, 100)
Pain	65	34.1 (29.1)	33 (0, 100)
Dyspnea	63	23.2 (28.5)	28 (0, 100)
Insomnia	64	29.7 (29.8)	30 (0, 100)
Appetite loss	65	24.6 (31.3)	31 (0, 100)
Constipation	63	20.1 (28.4)	28 (0, 100)
Diarrhea	63	31.2 (33.3)	33 (0, 100)
Financial difficulties	63	22.2 (30.5)	31 (0, 100)
Endocrine scale	62	20.4 (19.3)	17 (0, 67)
GI scale	62	32.2 (20.0)	27 (0, 87)
Treatment scale	37	21.6 (16.1)	22 (0, 56)
Social functioning scale	60	55.7 (21.2)	56 (0, 100)
Disease relates worries scale	38	48.0 (29.2)	44 (0, 100)
Muscle/bone pain symptom	60	33.9 (30.4)	33 (0, 100)
Sexual function	45	74.8 (33.5)	100 (0, 100)
Information/communication function	64	90.1 (20.3)	100 (0, 100)
Body image	63	19.6 (28.5)	0 (0, 100)

GI: Gastrointestinal; HR-QoL: Health-related quality of life; SD: Standard deviation.

**Table 6 jcm-11-01271-t006:** Questions patients missed in both questionnaires based on the perceptions of the patients.

Questions about Hair Loss
Medication-related questions
Questions about the psychological assistance from physicians during anti-tumoral treatment
Questions regarding surgical therapy
Questions about positive aspects regarding quality of life, selfcare for self and others, happiness
Questions on sports and work
Questions on loneliness and solitude

## Data Availability

The datasets during and/or analyzed during the current study are available from the corresponding author on reasonable request.

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
