# Peer review of "A Multinational Pilot Study on Patients’ Perceptions of Advanced Neuroendocrine Neoplasms on the EORTC QLQ-C30 and EORTC QLQ-GINET21 Questionnaires"

_jcm, 2022, doi:10.3390/jcm11051271_

Round 1

Reviewer 1 Report

It should be specified whether the study is prospective or previously conducted questionnaries were reviewed. It should be indicated whether the surveys regarding the questionnaries was carried out at the same time or not. If they were not carried out simultaneously, the time elapsed between both should be indicated.

There are some typographical mistakes in Table 3. In addition, the percentage of previous surgeries should be shown, since many of these impair the quality of life.

Quantitative variables should be expressed as mean or median depending on whether or not they fit a normal distribution. In Table 5 there are multiple items whose SD is greater than the mean, suggesting that they do not fit a normal distribution, so they should be expressed as median. This should be described in the Methods section.

Section 3.3.2. should probably not be so precise and a more generic description should be enough.

In section 3.3.3. statistical techniques should be used to determine whether the differences mentioned are significant. Their omission may mislead the reader. This should be described in the Methods section.

Author Response

We would like to thank the reviewer for the time spent reviewing the manuscript.

Point 1. It should be specified whether the study is prospective or previously conducted questionnaries were reviewed. It should be indicated whether the surveys regarding the questionnaries was carried out at the same time or not. If they were not carried out simultaneously, the time elapsed between both should be indicated.

Reply 1. Thanks to the reviewer to remark this point. As already mentioned in the manuscript this is a cross-sectional qualitative pilot study (line 92).  The study has been conducted prospectively. We have added this particular in the manuscript (lines 92 and 95).

Point 2. There are some typographical mistakes in Table 3. In addition, the percentage of previous surgeries should be shown, since many of these impair the quality of life.

Reply 2. Thanks to the reviewer for letting us see the typos in the table 3, we have revised accordingly. We agree with the reviewer, and we acknowledge that previous surgery may impair the quality of life. Since the focus was on the perception of patients on the EORTC QLQ-C30 and EORTC QLQ-G.I.NET21 we decided to focus primarily on current medical therapies which were mainly systemic. Reporting data on previous surgeries would have been appropriate.

Point 3. Quantitative variables should be expressed as mean or median depending on whether or not they fit a normal distribution. In Table 5 there are multiple items whose SD is greater than the mean, suggesting that they do not fit a normal distribution, so they should be expressed as median. This should be described in the Methods section.

Reply 3. Thanks to the reviewer who makes an accurate remark. We have now reported both the mean and median in Table 5.  In the methods’ section the following sentence was added: ‘the outcomes of the EORTC QLQ-C30 and EORTC QLQ-G.I.NET21 scores are presented in both mean and median scores to give insight in the distribution of the data’ (lines 159-160).

Point 4. Section 3.3.2. should probably not be so precise and a more generic description should be enough

Reply 4. Thanks to the reviewer for letting us see this point. We revised accordingly (lines 170-172).

Point 5. In section 3.3.3. statistical techniques should be used to determine whether the differences mentioned are significant. Their omission may mislead the reader. This should be described in the Methods section.

Reply 5. Thanks to the reviewer who makes this accurate remark. Solely descriptive statistics were applied to characterize the study population and therefore no statistical tests were used to determine differences.

Reviewer 2 Report

Your manuscript describes a very important problem regarding NET HR-QoL questionnaires. Due to the heterogeneity of localizations, hormonal secreting syndromes and proliferation indexes, clinical manifestations and impact in HR-QoL is very different from patients to patients. HR-QoL cancer questionnaires are designed in general for the most frequent malignant tumors and don`t fit so well in this rare and complex group of neoplasia. Even  EORTC QLQ- 113 G.I.NET21 is too generic and foccused mainly in carcinoid syndrome. So I think QoL questionaires in NETs must take into account not only   complaints generally related to tumor burden but also specific sub-questionaires according to the specific hormonal syndrome. 

I think this is the most important pitfall of the present manuscript. It should also mention this aspect and explore it in the discussion.

The other point I would like to emphasize, is about  NETTER 1 HR-QoL results, that I think should be discussed in more detail.

Author Response

We would like to thank the reviewer for the time spent reviewing the manuscript.

 Your manuscript describes a very important problem regarding NET HR-QoL questionnaires. Due to the heterogeneity of localizations, hormonal secreting syndromes and proliferation indexes, clinical manifestations and impact in HR-QoL is very different from patients to patients. HR-QoL cancer questionnaires are designed in general for the most frequent malignant tumors and don`t fit so well in this rare and complex group of neoplasia. Even EORTC QLQ- 113 G.I.NET21 is too generic and focused mainly on carcinoid syndrome. So, I think QoL questionnaires in NETs must take into account not only   complaints generally related to tumor burden but also specific sub-questionnaires according to the specific hormonal syndrome. 

Point 1. I think this is the most important pitfall of the present manuscript. It should also mention this aspect and explore it in the discussion

Reply 1. Thanks to the reviewer. We added a comment on that in the manuscript (lines 218-222, 332-338)

Point 2. The other point I would like to emphasize, is about NETTER 1 HR-QoL results, that I think should be discussed in more detail

Reply 2. Thanks to the reviewer. We added a comment on that in the manuscript (lines 225-233)

Reviewer 3 Report

If it is possible increase the sample. Questionaries analized patients treated in 2017

Author Response

We would like to thank the reviewer for the time spent reviewing the manuscript.

Point 1. If it is possible increase the sample. Questionaries analized patients treated in 2017.

Reply 1.

Thank you for your suggestions. We will consider to increase the sample for a further study.
